# Premorbid Personality Traits as Risk Factors for Behavioral Addictions: A Systematic Review of a Vulnerability Hypothesis

**DOI:** 10.3390/children10030467

**Published:** 2023-02-26

**Authors:** Daniela Smirni, Pietro Smirni, Gioacchino Lavanco, Barbara Caci

**Affiliations:** 1Department of Psychology, Educational Science and Human Movement, University of Palermo, 90128 Palermo, Italy; 2Department of Educational Sciences, University of Catania, 95124 Catania, Italy

**Keywords:** premorbid personality, neuroticism, internet gaming disorder, gambling, video games addiction, brain reward system

## Abstract

The debate on personality structure and behavioral addictions is an outstanding issue. According to some authors, behavioral addictions could arise from a premorbid personality, while for others, it could result from a pathological use of technological tools. The current study aims to investigate whether, in the latest literature, personality traits have been identified as predictors of behavioral addictions. A literature search was conducted under the PRISMA methodology, considering the most relevant studies of the five-factor model from the past 10 years. Overall, most studies on addiction, personality traits, and personality genetics proved that behavioral addiction may be an epiphenomenon of a pre-existing personality structure, and that it more easily occurs in vulnerable subjects with emotional instability, negative affects, and unsatisfactory relationships with themselves, others, and events. Such neurotic personality structure was common to any addictive behavior, and was the main risk factor for both substance and behavioral addictions. Therefore, in clinical and educational contexts, it becomes crucial to primarily focus on the vulnerability factors, at-risk personality traits, and protective and moderating traits such as extroversion, agreeableness, conscientiousness, and openness to experience; meanwhile, treatment of behavioral addictions is frequently focused on overt pathological behaviors.

## 1. Introduction

Over the past few decades, behavioral addictions have become a crucial issue for their growing epidemiological diffusion and for the relevant implications, both in health, and in family and educational contexts [1]. Therefore, a significant amount of the literature has shown a growing concern about the relationship between personality patterns and addictions [2,3,4,5,6,7], including a large number and broad range of non-chemical behavioral addictions, such as gambling, overeating, compulsive sex, exercise, compulsive shopping, video game playing, love, Internet use, and work [8,9,10,11,12,13,14,15]. 

Behavioral addictions, while not involving the use of drugs or chemicals, show the same features as substance addiction [16,17,18], such as repetitive behavior patterns, short-term reward, long-term personal and social problems, reduction or loss of frontal behavioral control, and high relapse rates [11,19,20,21]. They can arise as enjoyable leisure activities, especially in children and adolescents. However, over time, they can gradually remove individuals from daily activities and interpersonal relationships, and create a pathological, persistent, and uncontrollable need to continue and increase in quantity and frequency, without limits and control. Several studies have shown a possible continuum between frequent unproblematic habitual behaviors, overuse, and problematic and addictive behaviors [22,23,24,25,26,27]. Even the excessive use of social networking sites, for example, can easily become the salient activity, and require longer amounts of time, gradually leading to a vicious cycle of addiction [25] and assuming the operational characteristics of chemical and substance addiction, i.e., salience, mood modification, tolerance, withdrawal, conflict and relapse [28]. 

Additionally, in both chemical and behavioral addictions, similar genetic vulnerabilities and phenotypic associations between personality traits and cortical areas, especially prefrontal, have been suggested [29].

Therefore, the Diagnostic and Statistical Manual of Mental Disorders-5 edition (DSM-5) [30] modified the chapter on addictions from ‘substance-related disorders’ to ‘substance-related and addictive disorders’. However, it included only the gambling in this latter chapter, in relation to the similarities between gambling and substance addictions, while other behaviors, such as compulsive sexual behavior, compulsive shopping, problematic Internet use or theft were not included, as scientific evidence on the subject was not deemed sufficient. 

Conversely, the International Classification of Diseases-11th Revision (ICD-11) [31], which primarily targets issues of clinical utility, global applicability, and scientific validity, maintained the impulse control disorders as a single nosographic category [32]. Such a unique category includes behaviors with a global impact on public health, and many common clinical characteristics such as the repeated inability to resist an impulse, drive, or urge to perform a short-term rewarding act in the absence of substance use. Clinically, keeping these disorders in a single category, unlike the DSM-5 approach, should suggest that clinicians, who identify a given impulse control disorder, screen for other related impulse control disorders [32,33].

Therefore, to date, the debate on the relationship between personality patterns and behavioral addictions appears to be outstanding crucial issue. According to some authors, behavioral addictions could be the result of a pathological use of technological tools [34]. Conversely, according to others, they could arise from a premorbid personality structure; consequently, psychologically fragile subjects with lower self-esteem, introversion, low conscientiousness and agreeableness and high neuroticism, hyperactivity, high novelty seeking behavior, or low self-control and self-regulation can be easily involved in behavioral addictions [35,36,37,38,39,40].

In the last twenty years, in personality trait theory, a considerable number of studies used the Five-Factor Model (FFM) as a comprehensive taxonomy of personality traits, comprising behavioral, emotional, and cognitive factors [41]. In such a model, the five main traits have been thought of as enduring determinants of behavior throughout the lifespan and across different conditions, cultures, and countries [42,43], and have been labeled as openness, conscientiousness, extraversion, agreeableness, and neuroticism: openness to experience as the tendency to be curious, flexible, and open to novelty; conscientiousness as the tendency to modulate impulse control within the rules and ethical principles; extraversion as the tendency to establish satisfying interpersonal relationships in real life; agreeableness as the tendency to be tolerant, empathetic, and cooperative in interpersonal relations; neuroticism as the tendency to be emotionally unstable, and to perceive events negatively. Several studies showed high neuroticism, low conscientiousness, low agreeableness, and low extraversion in people with clinical symptoms, personality disorders, and substance use disorders [42].

However, some studies showed that the FFM does not actually provide comprehensive measures of personality, and developed alternative personality models [44], although researchers still predominantly assess the same five factors in studies on personality. The HEXACO model, for example, identified a six-factor model of personality: honesty-humility (H), emotionality (E), extraversion (X), agreeableness (A), conscientiousness (C), and openness to experience (O) [45,46]. Similarly, Supernumerary Personality Traits is an alternatively developed model of 10 traits not encompassed by the FFM: conventionality, seductiveness, manipulativeness, thriftiness, humorousness, integrity, femininity, religiosity, risk-taking and egotism [47]. A psychobiological model of personality identifies four traits of temperament (novelty seeking, harm avoidance, reward dependence, and persistence) and three character traits (cooperativeness, self-directedness, and self-transcendence) [48], derived from neurobiological, developmental, and genetic studies. Other models identify subclinical personality traits of Machiavellianism, psychopathy, narcissism, and everyday sadism [49], or the emotional dimension of intelligence as the ability to express, perceive, understand, and regulate emotions and social abilities [50,51].

The current study focused on the relationship between personality patterns and behavioral addiction, in order to investigate whether, in the literature of the last decade, an underlying personality structure has been identified that makes subjects more vulnerable to addiction. The mainstream personality psychology literature has gained increasing acceptance that the FFM represents a comprehensive taxonomy of personality for adults [52,53,54,55,56]. However, research on the FFM with children is also flourishing, demonstrating that personality traits predict school performance and intelligence [57,58,59], self-esteem [57], and measures of internalizing and externalizing disorders [57,60,61] in middle childhood and adolescence. Other studies on adolescents have analyzed the predictive role of personality traits for substance addiction such as drugs [62], but studies related to behavioral addiction adopted mainly a correlational design approach in studying the linear associations between personality traits defined in line with the FFM and Internet addiction [63], gaming addiction [64], or social networking addiction [65]. The current study aims to overlap this gap, starting from the consideration that the underlaying structure of personality for behavior, even if measured and empirically evaluated in adults, may be a feature of significant novelty and relevance, both to research and to the preventive policy of addictive behaviors, since the earliest ages. Identifying a premorbid personality pattern, if any, supporting the onset of addictive behavior, may have significant implications in research, in clinical, educational and youth policy contexts, and should be the ‘ubi consistam’ on which an effective program for preventing should be grounded from an early age [66]. Conversely, educational and rehabilitation programs, predominantly focused on controlling or modifying overt behavioral addictions, can be much more complex, especially when these behaviors are highly structured. Therefore, in such contexts, the current study may be accounted as an original contribution in the early prevention of non-chemical addiction. Although individual behavioral addictions may arise from an interaction between various personal, structural, and situational factors, such as biological vulnerability, psychological features, social environment, and the structural characteristics of the activity itself [67,68,69], addictive behaviors of any kind are likely to have many common features and probably also a general etiological matrix [11].

## 2. Materials and Methods

### 2.1. Design 

A systematic search strategy was used to identify relevant studies. This review com-plies with the PRISMA (Preferred Reporting Items for Systematic Reviews and Meta-Analyses) guidelines for the search, systematization, and reporting of systematic reviews [70]. It was not possible to conduct a meta-analysis because of the different kinds of studies considered in the current systematic review, which related to review and empirical studies, studies with different designs, or without a control group, and mixed outcome reporting.

### 2.2. Search Strategy

A literature search on the Ovid MEDLINE, Embase, PsycINFO, PubMed, Scopus (Elsevier), and Web of Sciences databases was conducted. The following keywords were used: behavioral addiction, personality traits, premorbid personality, neuroticism, Internet gaming disorder, gambling, video games addiction, vulnerability hypothesis, non-chemical addiction, coping, and brain reward system. The Boolean operators AND and OR were used. For the publication year, the period between 2011 and 2021 was selected, considering relevant studies from the last 10 years as an additional criterion. 

### 2.3. Study Selection

Three independent reviewers (P.S., D.S., B.C.) evaluated article eligibility. The progressive exclusion was performed by reading the abstract and, finally, the full text. 

The selection of studies was conducted systematically based on the following criteria: (a) written in English language; (b) behavioral addictions as a main topic; (c) publication between January 2011 and November 2021; (d) different types of studies likewise reviews, meta-analyses, case reports, randomized controlled studies, and observational studies on subjects with different behavioral addiction conditions; (e) follow a high methodological rigor; (f) published as journal articles, book chapters, and original manuscripts. Additionally, studies were excluded based on the following: (a) the Five-Factor Model was not used for the assessment of personality traits; (b) the full text could not be obtained; (c) there was a lack of transparency due to missing methodology information. Discrepancies regarding the inclusion/exclusion of studies were discussed within the group, and disagreements were resolved by consensus. 

### 2.4. Data Analysis

Data analysis was carried out by focusing on the kind of behavioral addiction that was focused by the reviewed studies, the personality trait model employed by authors, and the reported results. Three independent judges (P.S., D.S., B.C.) recorded the following data from the included articles: authors; year of publication; method and sample; results. The JBI Critical Appraisal Checklist for Systematic Reviews [71] was used to rate the quality of the included studies’ research evidence; it was an 11-item checklist used to rate the inclusion of each study with “yes”, “no”, or “unclear”, or “not applicable” in terms of the specific study characteristics. 

## 3. Results

A total of 108 original studies (see Figure 1) were obtained. After de-duplication and a topic relevance review of all the original studies, 105 studies were selected for further analysis. Finally, the studies where the five-factor model was not used for the assessment of personality traits, and manuscripts where the full text was not available, were excluded. From the remaining 65 studies, only 13 were further systematically classified and subjected to descriptive analysis after evaluating their ratings on the JBI Critical Appraisal Checklist for Systematic Reviews.

### 3.1. Personality Traits and Addictive Behaviors in the Reviews

Over the past decade, a few reviews and empirical studies, including about a hundred studies on more than 60,000 total subjects in different contexts and with different study designs and methodologies, reported almost unanimously that there were some significant relationships between addictive behaviors and personality patterns (Table 1). The studies considered used the Five-Factor Model as a standard measure of personality traits [72]. In general, all of the studies showed the same trend in demonstrating the involvement of personality traits in behavioral addiction, but also showed an evolution of the specific definition of the theoretical construct of behavioral addiction itself through the years.

Indeed, we found in Floros and Siomos [5] the initial definition of behavioral addiction as excessive Internet use (EIU), including only symptoms such as signs of using the Internet excessively to the detriment of their ‘real’ lives, to time spent with their families and friends, and to the ability to meet their obligations. This is coherent with the initial definition of Internet Addiction Disorder reported by Goldberg [73] and Young [74,75]. Successively, Kayiş et al. [4] adopted the definition of Internet Addiction Disorder, including indicators such as the inability to limit Internet use, continuing to use the Internet despite social or academic damage, and feeling anxious when Internet access is limited, but also Internet usage purposes such as gambling, chatting, playing games, and pornography. Finally, more coherently with the last DSM-V diagnostic approach, Gervasi and colleagues [5], and Şalvarlı and Griffiths [6] referred to Internet Gaming Disorder, focusing on the similarity between symptoms of excessive Internet gamers and those of people suffering from substance use disorder, such as playing compulsively; frequent and obsessive thoughts regarding the game to the exclusion of other interests; social isolation; psychological discomfort when gaming is reduced; reduction of social, recreational, work, educational, household, and/or other activities; disregard for one’s own and others’ needs because of the behavior; withdrawal when pulled away from gaming; and persistent and recurrent online activity resulting in clinically significant impairment or distress [5]. 

Specifically, Floros and Siomos [3], in examining 40 studies, found a positive correlation between Internet overuse and neuroticism, psychoticism, sensation/excitement seeking, and openness, and a negative correlation between extraversion, conscientiousness, agreeableness, reward dependence, and self-directedness. Moreover, the authors suggested a specific neurobiological substrate as a predisposition for addictive behaviors.

Very similar data were found in Kayiş et al. [4], who performed a meta-analysis that included 12 studies on a total sample of more than 12,000 individuals. All of the big five personality traits were significantly correlated with Internet addiction: neuroticism positively, and openness to experience, conscientiousness, extraversion and agreeableness negatively [76]. According to the authors, as individuals with high neuroticism are emotionally unstable and perceive events negatively, they can use online communication at the addiction level as a more comfortable relational condition. Conversely, conscientious subjects adequately control Internet use, since conscientiousness involves modulating impulse control within the rules. Likewise, agreeable individuals, being tolerant and empathetic, show a lower tendency to addictive Internet use. Since extraverted individuals interact positively with others, they can establish satisfying interpersonal relationships in real life, and can use the Internet at a relatively low level. Open individuals are curious, flexible, and open to novelty, and may prefer real-life environments to satisfy their interest rather than virtual ones.

A systematic review of Gervasi and colleagues [5], including 27 empirical studies on a total sample of 36,340 subjects, confirmed that high neuroticism, and low agreeableness, extraversion, conscientiousness, and openness to experience were associated with online gaming [77,78] and pathological Internet use [77,79,80]. More recently, in a review based on 21 empirical studies performed on a total sample of 16,536 gamers [6], high neuroticism, and low extroversion, conscientiousness, agreeableness, and openness to experience were also confirmed as risk factors for Internet gaming addiction.

Overall, therefore, the examined reviews support the profile of an Internet excessive user as an emotionally unstable subject driven by short-term gratification, with poor impulse control, and difficulty in managing their feelings and relationships.

### 3.2. Personality Traits and Individual Behavioral Addictions in the Empirical Studies

Several empirical studies showed significant relationships between specific personality traits and individual behavioral addictions (Table 2).

Andreassen and co-authors [81] examined seven different behavioral addictions (Facebook, video games, Internet, exercise, mobile phone, compulsive buying, and study addiction), in a sample of 218 university students (171 female) with a mean age of 20.7 years ± 3.0. Neuroticism and extraversion were positively associated with all of the seven behavioral addictions, while agreeableness was negatively correlated. 

Kuss et al. [82], in a cross-sectional online survey with 2257 English university students, found significantly higher neuroticism and low agreeableness associated to Internet addiction, while neuroticism was the strongest predictor factor. 

Similarly, Müller and colleagues investigated personality structure in different behavioral addictions (gambling, Internet and gaming), showing high neuroticism, low conscientiousness and low extraversion both in Internet gaming disorder and Internet addiction, while low conscientiousness was the strongest predictor [83,84]. According to the authors and to an etiopathological model [83], high neuroticism and introversion might motivate one to search for safe and comfortable environments of the Internet and online communication.

Wang and colleagues [85], in a cross-sectional study, investigated personality traits on three different online activities, Internet, gaming, and social networking, among a sample of 920 Chinese adolescents (583 females) (mean age sample 15.03 ± 1.59, females 14.98 ± 1.61, males 15.13 ± 1.56). Higher neuroticism was associated with internet and social networking addiction, but not with gaming addiction. Low openness was mildly significantly associated with gaming addiction, probably due to players with low openness tending to engage in gaming behavior instead of exploring new activities. Low conscientiousness was significantly associated with gaming addiction, but not with social networking addiction. Indeed, individuals with low conscientiousness may find the virtual gaming environment attractive. Extraversion was significantly associated with social networking addiction, while agreeableness was not related to any outcome measures.

Wittek et al. [86] examined a representative sample of 3389 gamers, aged 16–74 years (1351 females, mean age = 32.6 years), randomly selected from the National Population Registry of Norway. Video game addiction was positively associated with neuroticism, and negatively with conscientiousness. Subjects high on conscientiousness were three times less likely to belong to the group of addicted gamers, since they typically are dutiful and self-disciplined [42]; therefore, they may be incompatible with heavy video game playing. No significant relationship was found between extraversion or agreeableness and video game addiction.

An interesting study by Vaghefi and Qahri-Saremi [87], via two surveys from 275 participants who were social networking site-addicted, investigated not only the direct and moderate effects of three personality traits (neuroticism, conscientiousness, and agreeableness) but also their interaction through an analysis of the interaction effects. The study confirmed the positive effect of neuroticism and the negative effect of conscientiousness, while agreeableness was not directly associated with addiction. Furthermore, the authors found that the effect of each trait could be moderated by the other traits. Specifically, high neuroticism, which was directly related to addiction, also displayed an indirect effect on addiction by reducing the moderating effect of conscientiousness. That is, highly neurotic subjects lose the moderating influence of conscientiousness on addiction. Moreover, conscientiousness was negatively associated with addiction, but it also displayed an indirect effect on addiction by moderating the relation between agreeableness and addiction. That is, subjects with low conscientiousness and low agreeableness were more likely to develop addiction. In turn, subjects with low conscientiousness and high agreeableness displayed low addiction. 

Reyes et al. [88], in a large sample of Filipino gamers, found neuroticism to be positively correlated with pathological gaming, while the remaining Big Five personality traits were negatively correlated. They also found that conscientiousness was the strongest predictor of pathological gaming. 

Recently, Dieris-Hirche et al. [89] confirmed previous findings on general Internet addiction and personality traits in a cross-sectional study among a large sample of 820 video gamers (217 female) between the ages of 12 and 66 (M = 25.25 ± 10.31). High neuroticism and low conscientiousness were significant predictors for Internet gaming disorder. Problematic gamers showed significantly higher neuroticism as well as lower self-efficacy, extraversion, conscientiousness, and openness, while no significant difference was found in agreeableness. The largest effects were found for conscientiousness. According to the authors, excessive or chronic use of video gaming may result in a decrease in conscientiousness and, in turn, low conscientiousness may result in increased video gaming.

## 4. Discussion

Overall, the reviewed literature supported a putative personality profile of the behavioral addicted individual as an emotionally fragile and unstable subject, driven by short-term gratification, with poor impulse control and difficulty in managing their feelings and relationships. Moreover, the studies reported almost unanimously significant relationships between addictive behaviors and personality traits.

Therefore, it appears that the broad repertoire of potentially addictive activities can become particularly attractive to psychologically frail subjects. Barker and Kraut [90,91] proposed a ‘rich get richer’ model, according to which the Internet, for example, may consolidate or expand social networks to the subjects who are well-adjusted and socially successful. By contrast, poorly adjusted people may suffer deleterious effects from heavy Internet use, configuring a ‘poor get poorer’ condition. Therefore, by generalizing children and adolescents who find it difficult to deal with the complexity of real daily life, they would be particularly drawn to the various forms of online worlds to compensate for a lower psychosocial well-being and poor coping strategies. On the subject, Gentile et al. [92] suggested a bidirectional hypothesis. Children with psychological problems may be more attracted to video games (excitement hypothesis) and, in turn, find it less engaging to focus on activities requiring more control and sustained attention (displacement hypothesis). According to such hypotheses and to the operant conditioning model [93,94,95], online activities could become the psychological context for the structuring of addictive behaviors. 

### 4.1. Personality Traits and Behavioral Addiction 

In the examined studies, neurotic personality structure was common to any addictive behavior, and was the main risk factor for both substance and behavioral addictions. As individuals with high neuroticism are emotionally unstable and perceive events negatively, they can use online communication at the addiction level as a more comfortable relational condition. Moreover, personality studies have documented that neuroticism pervades every single dimension of the personality, and results in a negative and anxious perception of events, life, and oneself [96]. 

In neurotic individuals, cognitive performance, social expansion, and self-image appear to be conditioned by insecurity and low self-esteem, low confidence in others, uncertainty, and distress [97]. In such a personality framework, online worlds and digital socialization can significantly mitigate negative feelings in real life, social anxiety, and the stress of face-to-face communication, and ensure immediate rewards and a sense of belonging.

Conversely, conscientiousness has been correlated with addiction as a primary protective factor. Individuals with high conscientiousness engage and persist at goal-directed behaviors by modulating impulse control within the rules and in real life. Therefore, they responsibly control their impulses toward timeless use of technology and the online environment, and they are less likely to develop addictive behaviors.

Moreover, most studies have shown a protective role of agreeableness, as the tendency to be warm, friendly, sympathetic, and to avoid conflicts. Since people with low agreeableness are uncooperative and competitive when dealing with a conflict or difficult relationship, they can addictively use anonymous online environments, with few rules and little demand for friendly and cooperative behavior. 

It is interesting to point out that, according to Vaghefi and Qahri-Saremi [87], personality traits may interact with each other. Neuroticism can decrease the moderating effect of conscientiousness on addiction. That is, subjects high on neuroticism may display addictive behaviors despite good conscientiousness. Likewise, the effect of agreeableness can be moderated by conscientiousness. As the level of conscientiousness increases, the negative effect of agreeableness on addiction is reduced. Conversely, when conscientiousness decreases, the negative effect of agreeableness on addiction increases. 

The studies on openness to experience and extraversion were mixed and inconclusive [4]. Openness to experience refers to nonconformity, curiosity, and readiness to become involved in unknown experiences. Several studies found that openness to experience can be a protective factor against online gaming. However, individuals who are high on openness can be involved both in real and in virtual environments [4]. Indeed, some studies found a positive relationship with addiction [6], suggesting that players driven by immersion or socialization motives can be high on openness, and play to meet others and cooperate [98].

Other studies found no relationship or a negative relationship between openness to new experiences and pathological use of the Internet [99], suggesting that players low on openness to experience or driven by escaping, fantasy, or by a sense of achievement may be interested in the virtual world rather than experimenting with unknown new activities [2,83,85,88,89,100]. Openness, therefore, appears to be less relevant to virtual activities than all of the other Big Five personality traits.

Likewise, some studies reported a negative association between extraversion and Internet gaming disorder, supporting the protective effect of extraversion on addiction [96]. Extraverted individuals are friendly, sociable, and active; they interact positively with others, and establish satisfying interpersonal relationships in real life [76]. Therefore, they do not need to pathologically use the Internet to satisfy their need to belong [84]. At most, they can participate in the virtual environment to enhance social improvement according to ‘the rich get richer’ framework [90]. Conversely, introverted subjects are socially detached, insecure, and inhibited, and they can feel uncomfortable in face-to-face contacts; hence, they can use online virtual environments to create new identities and more rewarding social relationships [101], compensating for low life satisfaction, low self-esteem, and poor social skills [84], according to the social compensation model ‘the poor get richer’ [91]. Therefore, both extraverted and introverted individuals may be involved in gaming, but with different motives: the former, to enrich their social networks; the latter, to compensate for their poor and difficult social dimensions.

Overall, therefore, most studies, despite the diversity of methodologies, study designs, and the addictive behaviors examined, agree in considering the personality traits closely associated with addictive behaviors, supporting the assumption that a specific personality structure can be predisposed or be more vulnerable to any external stresses to addiction. High emotional instability and a neurotic personality structure, high introversion, and low levels of conscientiousness, agreeableness, and openness to experience, can represent the basis for an excessive and pathological use of the digital world where the confrontation with reality and the relationship with the other, the success, and the gratification appear simpler and easier to manage than in real life.

### 4.2. Heritability and Neuro-Anatomical Substrate of Personality

In this context, the long-standing debate on the inheritance of specific personality traits is of particular interest. Recent genetic studies support the assumption that some personality traits can be associated to genetic factors, with each contributing to the heritability of personality [3,29,102,103,104,105,106,107,108,109,110]. 

A recent meta-analysis [111] showed that, on average, about 40–49% of personality variance could be of genetic origin [112,113]. Several studies have reported that distinct genes are associated with specific cortical regions, and influence various behavioral traits, suggesting that the brain has a viable endophenotype linked to genes and behavior [29,106,114,115].

Recently, Riccelli et al. [116] investigated individual differences in personality traits in relation with local variabilities in surface area, cortical thickness, volume, and folding, in 507 young healthy adults (60% females) with a mean age of 29.2 years and an age range of 22–36 years, from the Human Connectome Project. They found that the five personality traits were associated to a neuro-anatomical substrate, especially in the prefrontal cortex. Namely, neuroticism was associated with a thicker cortex and smaller area, and folding in some prefrontal, temporal, and parietal regions. Higher extraversion was linked to higher thickness in the pre-cuneus and smaller superior temporal cortex and entorhinal area, and higher cortical folding in the fusiform gyrus. Openness was linked to a thinner cortex and greater area and folding in the parietal, temporal, and orbito-frontal regions. Agreeableness was correlated to a thinner orbito-prefrontal cortex, and a smaller fusiform gyrus and temporal areas. Conscientiousness was associated with a thicker cortex and smaller area, and folding in the fronto-temporal regions. 

Owens and colleagues [105] extended such findings by examining 1104 young healthy adults from the same Human Connectome Project, including monozygotic and dizygotic twins. They reported a significant phenotypic relationship between personality traits and cortical structures, especially the dorsolateral prefrontal cortex. Neuroticism, conscientiousness, and openness were the personality traits with the strongest neuro-anatomical representation. Valk et al. [29] confirmed previous studies supporting a relationship between personality and the frontal lobe [105,116,117]. Moreover, they suggested a genetic link between local structure in the frontal cortices and personality, showing that many of the phenotypic relationships between personality and local brain structure are driven, in part, by shared additive genetic effects rather than environmental factors alone. They found that a variance of between 30 and 57% (on average 42%) in personality traits was explained by additive genetic factors. 

### 4.3. Vulnerability Hypothesis

Therefore, genetic and cortical morphometry studies displayed a significant genetic dimension in personality structure, with an estimated heritability of around 50%, even with the limitations highlighted by some authors [118,119]. Therefore, a vulnerability hypothesis may be assumed, according to which pathological personality traits may be pre-existing risk factors that are influenced by genetic predisposition and environmental factors mutually and circularly interacting. Unfavorable environmental and family conditions can easily elicit addictive behaviors in vulnerable and susceptible subjects to any external pressures.

Overall, therefore, the literature proved that pathologically addicted subjects display a particular personality pattern that supports the pathological use of addiction activities, while excessive practice, in turn, increases pathological dimensions of the personality. 

## 5. Limitations

Some limitations of the current study should be considered. In the study, ‘addictive behavior’ was used as a comprehensive term encompassing the various conditions of non-chemical addiction, while each of the individual behavioral addictions show peculiar characteristics and should be approached in a differentiated way. Future research is needed to provide the relation between individual behavioral addiction and personality traits.

Moreover, there was no analysis of the measurement methods performed in the individual reviewed research studies to assess personality traits and behavioral addiction. Finally, the results of the research on excessive use were not differentiated from those on addiction. Further research is needed both to define the reliability of the various evaluation methods, and to differentiate between excessive use and overt addiction.

## 6. Conclusions

The recent literature on addiction, personality traits, and personality genetics supports the hypothesis of behavioral addiction as an epiphenomenon of a pre-existing personality structure. According to the empirical evidence discussed in the present study, behavioral addiction is more readily displayed in individuals with emotional instability, negative affects, and unsatisfactory relationships with themselves, others and events, according to the model of ‘the poor get poorer’.

In this context, the current study can be seen as an original contribution to the topic of early prevention in non-chemical addiction. Empirical evidence has amply demonstrated that individual behavioral addictions can result from an interaction between various personal, structural, and environmental factors. However, the literature data show that addictive behaviors have many common features probably attributable to premorbid personality traits that can be considered a general etiological matrix. Therefore, an effective early prevention program for non-chemical addictions in children and adolescents must include interventions that promote healthy emotional and personological development.

The literature data, therefore, support relevant practical implications. While the treatment of addictive behaviors frequently focuses on overt pathological behaviors, the literature suggests early intervention on the premorbid personality structures. Clinical practice, education, and youth policy should focus on the vulnerability factors underlying behavioral addictions, on the early identification of children and adolescents with at-risk personality patterns, and on increasing protective traits such as extroversion, agreeableness, conscientiousness, and openness to experience. An emphasis on the preventive dimension appears even more relevant for children and adolescents. For any educational and rehabilitation programs focusing primarily on overt and highly structured addictive behavior, the outcomes may be much more modest when implemented later compared to early prevention programs.

## Figures and Tables

**Figure 1 children-10-00467-f001:**
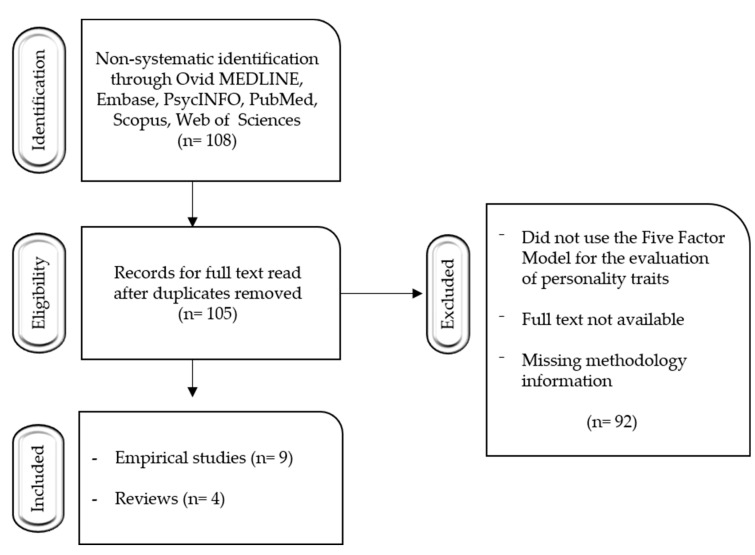
Selection of studies flowchart.

**Table 1 children-10-00467-t001:** Personality traits and addictive behaviors in the systematic reviews.

Review	Studies Sample	Theorethical Definition of Behavioral Addiction	Personality Model	Main Results
Floros and Siomos 2014 [3]	40 studies Sample not reported	Excessive Internet use	FFM Eysenck’s PEN model Cloninger’s psychobiological model Zuckerman’s alternative FFM Cattell’s 16 personality factors	Positive correlations with neuroticism, emotional stability, psychoticism, sensation/excitement seeking, and openness. Negative correlations with extraversion, conscientiousness, agreeableness, reward dependence, and self-directedness.
Kayiş et al., 2016 [4]	12 studies n = 12019	Internet addiction	FFM	Positive correlation with neuroticism, negative correlations with openness, conscientiousness, extraversion, and agreeableness.
Gervasi et al., 2017 [5]	27 empirical studies n = 36,340	Online gamers and Internet addicted	FFM	High neuroticism, low agreeableness, extraversion, conscientiousness, and openness
Şalvarlı and Griffiths, 2019 [6]	21 empirical studies n = 16,536	Internet gaming addiction	FFM	High neuroticism, low extroversion, conscientiousness, agreeableness, and openness. Some studies found a positive relationship with neuroticism; others found no relationship. Extraversion was negatively correlated in some studies; in others, it was unrelated. Conflicting results on the relationship with conscientiousness and agreeableness; negative association or no relationship with openness.

Legend: FFM = Five-Factor Model.

**Table 2 children-10-00467-t002:** Personality traits and individual behavioral addictions.

Study	Sample	Addiction Behavior	Personality Traits
Andreassen et al., 2013 [81]	218 university students (171 female), mean age 20.7 years ± 3.0	Facebook, video game, Internet, exercise, mobile phone, compulsive buying, and study addiction	Neuroticism and extraversion positively associated with all the seven behavioral addictions, agreeablenes negatively correlated. Openness negatively associated with Facebook and mobile phone; conscientiousness negatively related to Facebook, video gaming, Internet, buying, and positively to exercise and studying.
Kuss et al. (2013) [82]	2257 university students (1.438 females) mean age 22.67 ± 6.34 range 18–64	Internet gaming	Higher neuroticism and low agreeableness associated to Internet addiction; neuroticism was the strongest predictor factor.
Müller et al., 2013 [83]	70 male patients mean age 29.3 ± 10.66 range 16–64	Internet addiction	Low conscientiousness as a risk factor, highest scores in neuroticism in patients with Internet addiction and depression.
Müller et al., 2014 [84]	115 internet gaming patients (mean age 22.9 years ± 6.13) 122 pathological gambling (mean age 32.3 years ± 11.53) 167 control subjects mean age 21.0 years ± 6.48	Internet gaming gambling control group	High neuroticism, low conscientiousness, and low extraversion were characteristics of Internet gaming disorder, low conscientiousness was the strongest predictor. Internet gaming patients displayed lower extraversion than all the other groups, including pathological gamblers; gamblers were characterized by elevated extraversion.
Wang et al. 2015 [85]	920 Chinese adolescents (583 females) mean age sample 15.03 ± 1.59 females 14.98 ± 1.61 males 15.13 ± 1.56	Internet use, gaming, and social networking	Higher neuroticism associated with Internet and social networking addiction but not gaming addiction. Low openness mildly significantly associated with gaming. Low conscientiousness significantly associated with gaming, but not social networking. Extraversion significantly associated with social networking; agreeableness was not related to any outcome measures.
Wittek et al. (2016) [86]	3389 gamers (1.351 females) aged 16–74 years mean age = 32.6 years	Video gaming	Video game addiction positively associated with neuroticism and negatively with conscientiousness. No relationship with extraversion and agreeableness.
Vaghefi and Qahri-Saremi (2018) [87]	275 students (51% women) mean age 21 ± 5.99 range 18–39	Social networking sites	Positive effect of neuroticism and negative effect of conscientiousness, while agreeableness was not associated with addiction. High neuroticism displayed an indirect effect on addiction by reducing the moderating effect of conscientiousness. Conscientiousness was negatively associated with addiction, but it also displayed an indirect effect on addiction by moderating the relation between agreeableness and addiction.
Reyes et al. (2019) [88]	1026 (491 females) mean age 23.57 ± 4.72	Gaming disorder	Neuroticism positively correlated with pathological gaming, the remaining Big Five personality traits were negatively correlated. Conscientiousness was the strongest predictor of pathological gaming.
Dieris-Hirche et al. (2020) [89]	820 video gamers (217 female) between the ages 12 and 66 (M = 25.25 ± 10.31).	Internet gaming disorder	High neuroticism and low conscientiousness were significant predictors for Internet gaming disorder. Problematic gamers showed significantly higher neuroticism and lower self-efficacy, extraversion, conscientiousness, and openness; no significant difference was found in agreeableness. The largest effects were found for conscientiousness.

## Data Availability

Not applicable.

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
