# Peer review of "Premorbid Personality Traits as Risk Factors for Behavioral Addictions: A Systematic Review of a Vulnerability Hypothesis"

_children, 2023, doi:10.3390/children10030467_

Round 1

Reviewer 1 Report (Previous Reviewer 2)

Thank you for the opportunity to review the manuscript.

The work submitted for review examines a topic of great relevance in the field of psychology. The topic is very important, and the conceptual analysis made in the text is quite deep. The literature consulted is quite current and the number of articles consulted is adequate. The methodology of the study is well explained, with emphasis on search engines and databases. I would like to thank the efforts by the authors of the manuscript and congratulate them on the work. Overall, the writing is clear, the goals are well described, the introduction should explain the objectives of the study based on the review of the previous literature and the conclusions are properly made and presented. I consider that the constructs proposed in the abstract of the work are quite well explained. Therefore, the manuscript brings significant knowledge of the scientific literature so and still covers existing gaps in the field. On a formal level, the manuscript complies with the requirements of the Journal and references are written in accordance with the regulations of the Journal. The work is ambitious, and the results confirm most of the hypotheses and the relevance and potential of the work is therefore recognized, but this Reviewer considers that minor changes are needed to the manuscript is publishable. In this sense, it should better explain the novelty and relevance of the work considering the previous empirical evidence and should better describe the practical implications. Finally, I wish the Authors the best in continuing this line of research.

 Best wishes for Authors.

Author Response

We are truly honored by these positive comments and we thank the reviewer for his/her words. Following his/her suggestion, we have reiterated what are the novelty of the article, taking up what was previously specified in the aim of the study (last lines of the introductory session, lines 129-135) also in conclusion, describing better the practical implications.

Reviewer 2 Report (New Reviewer)

Dear Authors

The topic that You cover seems to be of high importnace. However, there are some possible corrections that You should take into consideration.

You have high number of searching key words, what to some extent, make Your articles' main concern difficult to understrand. It also provokes the high number of references what is also not easy to read. Maybe You should excluede some of them.

Your idea is interesting, especially because of the high number of different addictions occurance, but according to me You should concentrate more on chosen aspects.

The title of Your article is correct but too general. 

Nonetheless, I consider Your articele as highly important and interesting in the area.

Best regards

Author Response

The topic that You cover seems to be of high importance. However, there are some possible corrections that You should take into consideration.

You have high number of searching key words, what to some extent, make Your articles' main concern difficult to understand. It also provokes the high number of references what is also not easy to read.

AR: We thank the reviewer for the positive comment. We agree that we have a high number of searching key words, following your suggestion we have reduced them to make easier to understand the main concern. As a result, some references have been excluded to make them more readable.

Maybe You should exclude some of them.

AR: Following suggestion, we have excluded some of them.

Your idea is interesting, especially because of the high number of different addictions occurrence, but according to me You should concentrate more on chosen aspects.

AR: Thanks for the positive comment, we think we covered high number of different addictions for the article type. We hope that having eliminated some references and better specified the main aspects of the article, it will be clearer that it can be seen as an original contribution to the topic of early prevention in non-chemical addictions, which has very important practical implications in the clinical setting.

The title of Your article is correct but too general. 

AR: We thank the reviewer for the suggestion, we have edited accordingly

Nonetheless, I consider Your article as highly important and interesting in the area.

AR: Thanks again for the positive comments and encouragement

This manuscript is a resubmission of an earlier submission. The following is a list of the peer review reports and author responses from that submission.

Round 1

Reviewer 1 Report

The manuscript examined "Premorbid personality traits and behavioral addictions in children and adolescents". Unfortunately, this aim is not realized in this study.

First of all, only 2 reviewed publications regarded adolescents (anyone among children), and one of them did not consider classical Big Five traits (namely, González-Bueso et al., 2020).  The rest of reviewed works included adult or older participants, so comparing these effects with the adolescent samples is highly inappropriate.

Secondly, nothing is known about the measurement methods performed to assess personality traits and "behavioral addiction". Some publications do not regard even addiction, but "excessive use", so the results are different from those about "addiction", and cannot be combined with them. Also, personality traits can be measured in several methods, including longer and shorter scales (e.g., some include 120 items, the other 10), some are more reliable and the others less or not all, therefore the results of these studies is also inappropriate.

Author Response

The manuscript examined "Premorbid personality traits and behavioral addictions in children and adolescents". Unfortunately, this aim is not realized in this study.

R: We agree with the reviewer. Our aim was not realized in the previous version of the manuscript. We resized the goal and also changed the title following the reviewer’s suggestions.

First of all, only 2 reviewed publications regarded adolescents (anyone among children), and one of them did not consider classical Big Five traits (namely, González-Bueso et al., 2020).  The rest of reviewed works included adult or older participants, so comparing these effects with the adolescent samples is highly inappropriate.

R: We agree with the reviewer that it is inappropriate to compare data regarding adult or older participants with the adolescent samples. However, the aim of the work was to investigate whether personality traits as predictors of behavioral addictions had been identified in the recent literature. Based on such empirical evidence, we wanted to emphasize that a preventive policy from an early age can be an element of relevance and great utility both in the clinical and research fields. We hope we have clarified this in the revised version of the work.

Moreover, the work cited by the reviewer had been incorrectly inserted and now removed from the review.

Secondly, nothing is known about the measurement methods performed to assess personality traits and "behavioral addiction". Some publications do not regard even addiction, but "excessive use", so the results are different from those about "addiction", and cannot be combined with them. Also, personality traits can be measured in several methods, including longer and shorter scales (e.g., some include 120 items, the other 10), some are more reliable and the others less or not all, therefore the results of these studies is also inappropriate.

R: We thank the reviewer for these comments. We believe that these highlighted aspects are in fact the limitations of our review. For this we have expanded the limitations accordingly.

Reviewer 2 Report

Firstly, thank you for the opportunity to review the manuscript. The work submitted for review examines a topic of great relevance in the field of psychology. The topic is very important, and the conceptual analysis made in the text is quite deep. The literature consulted is quite current. I would like to thank the efforts by the authors of the manuscript and congratulate them on the work. Overall, the writing is clear, the goals are well described, the introduction should explain the objectives of the study based on the review of the previous literature and the conclusions are properly made and presented. I consider that the constructs proposed in the abstract of the work are quite well explained. Therefore, the manuscript brings significant knowledge of the scientific literature so and still covers existing gaps in the field. On a formal level, the manuscript complies with the requirements of the Journal and references must be written in accordance with the regulations of the Journal. The work is ambitious, and the results confirm most of the hypotheses and the relevance and potential of the work is therefore recognized. In this sense, it should better explain the novelty and relevance of the work considering the previous empirical evidence and should better describe the practical implications. In addition, the limitations section should be better described. Finally, I wish the Authors the best in continuing this line of research.

Best wishes for Authors.

Author Response

Firstly, thank you for the opportunity to review the manuscript. The work submitted for review examines a topic of great relevance in the field of psychology. The topic is very important, and the conceptual analysis made in the text is quite deep. The literature consulted is quite current. I would like to thank the efforts by the authors of the manuscript and congratulate them on the work. Overall, the writing is clear, the goals are well described, the introduction should explain the objectives of the study based on the review of the previous literature and the conclusions are properly made and presented. I consider that the constructs proposed in the abstract of the work are quite well explained. Therefore, the manuscript brings significant knowledge of the scientific literature so and still covers existing gaps in the field. On a formal level, the manuscript complies with the requirements of the Journal and references must be written in accordance with the regulations of the Journal. The work is ambitious, and the results confirm most of the hypotheses and the relevance and potential of the work is therefore recognized. In this sense, it should better explain the novelty and relevance of the work considering the previous empirical evidence and should better describe the practical implications. In addition, the limitations section should be better described. Finally, I wish the Authors the best in continuing this line of research.

R: We thank the reviewer for these positive comments that encourage us to continue research in this area.

Following her/his suggestions we have better explained the novelty and relevance of the work considering previous empirical evidence and described practical implications both in the clinical and research field.

Thank you very much for these comments that help us to continue such hard and complex work as that of the researcher.

Round 2

Reviewer 1 Report

As I previously showed, the results of this work are unreliable, because the authors did not show what methods assessed personality traits, and (in particular) behavioral addictions. This work is not sound science. The authors mixed internet addiction (which is not clinically confirmed) with gaming addiction. There is no clearly stated and not defined in the introduction, so I am not sure whether the authors see differences between gaming addiction and Internet addiction. This is also too general, since people use the Internet for different motivations, and these various populations show distinctive behavior. I cannot agree to generalize all internet-related behaviors to one data set. This is out of scientific standards. I suggest selecting gaming addiction for the study as only one justified clinical disorder.

The method to assess individual differences in behavioral addiction and personality traits also depends on theory (which was not introduced exhaustively). Differences between distinct approaches were not sufficiently described, for example between Costa and MacCrae's and Goldberg's approaches to Big-Five personality. Still, the authors seem not to consider measurement reliability an important question for such a "review". It should be clearly stated the inclusion and exclusion criteria are based on the selective measurement tools and the high reliability of previously published works. Also, all measurement methods presented in reviewed studies should be described in the Introduction. Table 2 should include the name of each measure, with the number of items and reliability coefficient for each scale of personality and gaming addiction.

Author Response

1) As I previously showed, the results of this work are unreliable, because the authors did not show what methods assessed personality traits, and (in particular) behavioral addictions. This work is not sound science. The authors mixed internet addiction (which is not clinically confirmed) with gaming addiction. There is no clearly stated and not defined in the introduction, so I am not sure whether the authors see differences between gaming addiction and Internet addiction. This is also too general, since people use the Internet for different motivations, and these various populations show distinctive behavior. I cannot agree to generalize all internet-related behaviors to one data set. This is out of scientific standards. I suggest selecting gaming addiction for the study as only one justified clinical disorder.

R: The reviewer’s comment does not appear proper with respect to the purposes and characteristics of our submitted manuscript.

In the introduction it is clearly stated (final lines) that the work aims to investigate whether the literature has identified personality traits that make particular individuals more available than others to addictive behaviors of any kind (for example, gaming, gambling, overeating, shopping, internet, work etc.).

The question is: is there a sort of common denominator in the personality structure that favors the appearance of behaviors with the characteristics described by Griffiths: salience, mood modification, tolerance, withdrawal, relapse and conflict?

In other words, the purpose of the work was not the study of single forms of behavioral addictions and personality structure. It is obvious that each single form of addiction can have a specific single matrix, but all behavioral addictions likely have a common one.

(line 120 and following) The current study has focused on the relationship between personality patterns and behavioral addiction in order to investigate whether, in the literature of the last decade, an underlying personality structure has been identified that makes subjects more vulnerable to addiction.

(line 131 and following) Therefore, in such context, the current study may be accounted as an original contribution in the early prevention of non-chemical addiction. Although individual behavioral addictions may arise from an interaction between various personal, structural, and situational factors, such as biological vulnerability, psychological features, social environment, and the structural characteristics of the activity itself, however, addictive behaviors of any kind are likely to have many common features and probably also a general etiological matrix.

 (from line 281) Overall, the reviewed literature supported a personality profile of the behavioral addicted individual as an emotionally fragile and unstable subject, driven by short-term gratification, with poor impulse control and difficulty in managing their feelings and relationships. Moreover, the studies have reported almost unanimously significant relationships between addictional behaviors and personality traits. It would therefore appear that the broad repertoire of potentially addictive activities can become particularly attractive to psychologically frail subjects.

(Line 434 and following) In the study, 'addictional behavior' was used as a comprehensive term encompassing the various conditions of non-chemical addiction as a whole, while each of the individual behavioral addictions show peculiar characteristics and should be approached in a differentiated way. Future research is needed to provide the relation between individual behavioral addiction and personality traits.

2) 

The method to assess individual differences in behavioral addiction and personality traits also depends on theory (which was not introduced exhaustively). Differences between distinct approaches were not sufficiently described, for example between Costa and MacCrae's and Goldberg's approaches to Big-Five personality. Still, the authors seem not to consider measurement reliability an important question for such a "review". It should be clearly stated the inclusion and exclusion criteria are based on the selective measurement tools and the high reliability of previously published works. Also, all measurement methods presented in reviewed studies should be described in the Introduction. Table 2 should include the name of each measure, with the number of items and reliability coefficient for each scale of personality and gaming addiction.

R: While sharing the methodological observations of the reviewer, it should be noted that the works that support the conclusions of our manuscript are works published in peer-reviewed scientific journals.

Taken together, the reviewed studies document the existence of a relationship between personality structure and various forms of behavioral addiction.

These data seemed sufficient to support the thesis that there are personality traits that can make a subject more vulnerable to addiction.

It was not part of the aim of the work describe, for example,

…the differences between distinct approaches…, for example between Costa and MacCrae's and Goldberg's approaches to Big-Five personality.